# A Hierarchical Multi-Feature Point Cloud Lithology Identification Method Based on Feature-Preserved Compressive Sampling (FPCS)

**DOI:** 10.3390/s25175549

**Published:** 2025-09-05

**Authors:** Xiaolei Duan, Ran Jing, Yanlin Shao, Yuangang Liu, Binqing Gan, Peijin Li, Longfan Li

**Affiliations:** School of Geosciences, Yangtze University, Wuhan 430100, China; 2023720556@yangtzeu.edu.cn (X.D.); 500171@yangtzeu.edu.cn (Y.S.); liuygis@foxmail.com (Y.L.); 2023720554@yangtzeu.edu.cn (B.G.); 2023710527@yangtzeu.edu.cn (P.L.); 2022000381@yangtzeu.edu.cn (L.L.)

**Keywords:** laser point cloud, lithology identification, random forest, machine learning

## Abstract

Lithology identification is a critical technology for geological resource exploration and engineering safety assessment. However, traditional methods suffer from insufficient feature representation and low classification accuracy due to challenges such as weathering, vegetation cover, and spectral overlap in complex sedimentary rock regions. This study proposes a hierarchical multi-feature random forest algorithm based on Feature-Preserved Compressive Sampling (FPCS). Using 3D laser point cloud data from the Manas River outcrop in the southern margin of the Junggar Basin as the test area, we integrate graph signal processing and multi-scale feature fusion to construct a high-precision lithology identification model. The FPCS method establishes a geologically adaptive graph model constrained by geodesic distance and gradient-sensitive weighting, employing a three-tier graph filter bank (low-pass, band-pass, and high-pass) to extract macroscopic morphology, interface gradients, and microscopic fracture features of rock layers. A dynamic gated fusion mechanism optimizes multi-level feature weights, significantly improving identification accuracy in lithological transition zones. Experimental results on five million test samples demonstrate an overall accuracy (OA) of 95.6% and a mean accuracy (mAcc) of 94.3%, representing improvements of 36.1% and 20.5%, respectively, over the PointNet model. These findings confirm the robust engineering applicability of the FPCS-based hierarchical multi-feature approach for point cloud lithology identification.

## 1. Introduction

Lithology identification, as a pivotal technique for geological classification and interpretation, holds significant scientific value in fields such as resource exploration, geological hazard prevention, and engineering safety [1]. However, traditional lithology identification methods face limitations in application scenarios due to challenges such as complex stratigraphic structures, insufficient representativeness of core sampling, and low resolution of geophysical exploration techniques [2]. With the rapid development of 3D laser scanning technology [3], outcrop point cloud data acquisition has substantially enhanced the precision, efficiency, and 3D characterization capabilities of lithology identification, driving innovation in geological engineering applications.

For outcrop point cloud lithology identification, manual visual classification achieves the highest accuracy [4] but becomes impractical when handling point cloud interpretation tasks at scales of tens of millions [5,6]. Consequently, researchers have turned to machine learning-based approaches [7], forming two primary technical pathways: unsupervised and supervised methods [8,9,10]. Unsupervised lithology identification achieves autonomous data stratification through point cloud clustering [11] and feature auto-encoding [12], such as K-means [13] and spectral clustering algorithms [14]. These methods offer advantages of prior knowledge independence and high algorithmic efficiency [15,16]. However, their performance is often hindered by isotropic assumptions, rock heterogeneity, and computational complexity bottlenecks [17,18]. Although improvements have been attempted—such as density-based spatial segmentation [19] and geometrically constrained hierarchical clustering [20]—these methods remain overly sensitive to similarity metrics, leading to low accuracy in identifying complex lithological boundaries. Early supervised methods adopted a “feature engineering + classifier” paradigm, such as SVM models based on normal vector histograms [21] and random forest algorithms integrating multi-scale curvature features [22]. While outperforming unsupervised methods, they heavily rely on manually designed feature extraction pipelines (e.g., normal vector histograms, multi-scale curvature descriptors), which require significant computational resources for intermediate processing steps such as neighborhood searching, gradient calculation, and dimensionality reduction. This dependency not only introduces algorithmic bottlenecks but also limits generalization to complex geological structures where handcrafted features may fail to capture critical discriminative patterns. For instance, curvature-based features exhibit higher accuracy in identifying sedimentary bedding than massive igneous rocks [23,24,25]. This highlights the challenges faced by early supervised methods, including poor model generalizability due to handcrafted features and the curse of dimensionality from high-dimensional features [26].

Deep learning methods, capable of autonomously generating high-dimensional features [27,28,29], have overcome the limitations of traditional approaches. PointNet [30,31,32] pioneered direct processing of unstructured point clouds by extracting global features via max-pooling layers [33]. PointNet++ [34] introduced hierarchical sampling to significantly improve fine-grained classification accuracy [35,36], while the Dynamic Graph Convolutional Neural Network (DGCNN) enhanced complex point cloud representation through dynamic neighborhood updates [37,38,39,40]. These deep architectures demonstrate strong potential for outcrop point cloud lithology identification [41,42]. However, existing deep learning models demand large labeled datasets and incur high training costs [43,44]. Recent studies on feature optimization of point clouds [45] and 3D geological modeling techniques [46] further validate the necessity of multi-scale feature fusion for complex lithology identification. Addressing these challenges, this study proposes an improved feature selection mechanism for random forest classifiers, integrating Feature-Preserved Compressive Sampling (FPCS) and a Multi-Level Sampling (MLS) architecture. This approach maintains lithology identification accuracy while reducing reliance on training data, offering an optimized solution for practical applications. Additionally, PointNet’s tendency to overfit high-frequency lithologies further highlights its class imbalance issues.

## 2. Methodology

This study proposes a lithology identification framework for 3D outcrop point clouds, comprising three feature extraction modules: Feature-Preserved Compressive Sampling (FPCS), Multi-Level Sampling (MLS), and hierarchical feature fusion, along with a random forest-based lithology classification module. The framework operates through four sequential stages (Figure 1):

### 2.1. FPCS-Based Point Cloud Resampling

The Feature-Preserved Compressive Sampling (FPCS) method is grounded in the principles of graph signal processing, aiming to reduce the volume of point cloud data while retaining critical geospatial information. The core of this approach lies in utilizing graph filters to extract essential features from point clouds, such as edges, keypoints, and flatness. Subsequently, an optimal resampling distribution is derived by minimizing reconstruction errors, where the optimal sampling probability *π_i_*^*^ for each point is proportional to the L2-norm of its multi-scale feature vector ∥*fi*(*X*)∥_2_ (Equation (7)), meaning points with larger feature magnitudes are prioritized proportionally. The FPCS method enhances the efficiency, accuracy, and robustness of lithology identification by integrating three key processes: graph topology modeling to characterize spatial–geometric relationships, multi-scale feature extraction via a graph filter bank (low-pass, band-pass, and high-pass), and resampling distribution optimization to suppress noise interference.

#### 2.1.1. Graph Topology Feature Modeling

The FPCS algorithm effectively preserves critical structural features of complex geological formations, with its adaptive graph model structure accurately capturing the spatial–geometric relationships and multi-source attribute correlations within point cloud data. When characterizing the topological structure of heterogeneous rock layer surfaces, traditional Euclidean distance metrics inadequately capture complex geological structures (e.g., fractures/folds). Thus, we compute geodesic distance *d_g_*(*i*,*j*) via Dijkstra’s algorithm (Figure 2), with adjacency weights *W_i,j_* integrating dual constraints.

In the figure above, the core of the dual-constrained adjacency matrix optimization framework is the adjacency matrix W, which is designed to sparsify the spatial associations among nodes in the point cloud, as shown in Equation (1).(1)$$W_{i,j}=exp\left(-\frac{{\left[d_{g}(i,j)\right]}^{2}}{2\sigma^{2}}\right)\cdot \frac{1}{1+\kappa |\nabla z|}$$

The element *W_i,j_* represents the connection weight between point *i* and point *j*, where a higher weight indicates stronger geometric attribute correlations between the two points. The model incorporates two critical parameters, *σ* and ∇*z*. The parameter *σ* controls the neighborhood radius to capture local geometric similarity. The value of *σ* is determined by the point cloud density and the thickness of the target rock layer, which can be optimized through statistical analysis of local curvature to avoid over-smoothing or under-connection. The elevation gradient ∇*z* (calculated as the standard deviation of *z*-coordinates within the neighborhood), combined with the sensitivity coefficient *κ*, suppresses cross-layer connections. When adjacent nodes are located on different lithological interfaces (e.g., fault planes), the gradient value increases significantly, thereby reducing the connection weight and preventing feature confusion due to cross-layer interference.

To comprehensively characterize lithological information, a multi-dimensional heterogeneous feature space is constructed:(2)$$f_{i}={\left[x_{i},y_{i},z_{i},\rho_{i},\lambda_{i}^{\left(1\right)},\ldots ,\lambda_{i}^{\left(k\right)}\right]}^{T}$$

In the equation, *x*, *y*, *z* are geometric features used to describe the macroscopic morphology of the rock mass. Principal Component Analysis (PCA) is applied to compute the local covariance matrix, extracting derived features such as normal vectors and curvature. *ρ* denotes the reflectance intensity, reflecting the material properties of the rock surface and closely related to mineral composition. *λ* represents multispectral features, where *k* band information distinguishes lithological spectral response variations. The feature vectors undergo max-min normalization to eliminate dimensional discrepancies, and Mahalanobis distance is adopted to measure feature similarity, accounting for inter-dimensional correlations. Concurrently, a sparse graph topology is constructed to enhance computational efficiency: a KD-tree-based fast retrieval of *k*-nearest neighbors (*k* = 15∼30) is performed, retaining only significant connections (*W_i_*_,*j*_ > *ε*) to sparsely store the adjacency matrix. This provides a structured foundation for subsequent multi-scale filtering. The geodesic distance preserves the continuity of rock layers, gradient-sensitive terms enhance bedding plane recognition, and the multi-dimensional feature space integrates geometric and physical property parameters, collectively ensuring a hierarchical extraction of lithological features. To mitigate computational overhead in geodesic distance calculation, we implement two optimizations:(1)Neighborhood Restriction: Dijkstra’s algorithm is constrained to local neighborhoods (*k* = 30 nearest points) via KD-tree acceleration, reducing complexity from *O*(*N*^2^) to *O*(*NlogN*).(2)Parallel Batch Processing: Disjoint point clusters are processed concurrently using OpenMP (Figure 2), leveraging multicore CPU architectures.

#### 2.1.2. Multi-Scale Feature Extraction Based on Graph Filter Banks

Graph topology modeling achieves structured representation of input point clouds. To extract multi-scale lithological features, this study proposes a multi-scale feature characterization method based on graph filter banks. The methodology constructs a three-tier filter bank comprising low-pass, band-pass, and high-pass filters, which, respectively, extract lithological features at macro-, meso-, and micro-scales. This approach ensures effective data denoising while preserving the geometric structures and inherent attributes of outcrop surfaces. The structure of this three-tier filter bank and its correlation with geological features are illustrated in Figure 3. The low-pass filter layer suppresses high-frequency noise via a Gaussian kernel filter and applies the graph Laplacian operator Δ to perform smoothing operations on the point cloud data, as demonstrated in Equation (3).(3)$$L_{low}=e^{-\tau \Delta }$$

In the equation, the parameter *τ* controls the smoothing intensity, achieving a balance between noise reduction and detail preservation. The low-pass filter layer not only eliminates high-frequency noise within the point cloud but also preserves the integrity of large-scale geometric structures in the outcrop. The smoothing effect of the low-pass filter and its preservation of macro-scale continuity are shown in Figure 3 (left panel). In the band-pass filter layer, Haar wavelet basis functions are adopted to detect lithological abrupt interfaces and extract mid-frequency features, as formalized in Equation (4).(4)$$\psi_{j,k}=\frac{1}{\sqrt{2^{j}}}\left(\phi \left(\frac{t-k2^{j}}{2^{j}}\right)-\phi \left(\frac{t-\left(k+1\right)2^{j}}{2^{j}}\right)\right)$$

In the equation, *ψ_j_*_,*k*_ represents the Haar wavelet basis function, where *j* is the decomposition scale parameter and *k* denotes the translation parameter. By adjusting *j* and *k*, multi-scale band-pass wavelet decomposition effectively identifies key mid-frequency signals such as rock layer joints and fractures, precisely detects lithological abrupt interfaces at varying depths, and filters out high-frequency noise and low-frequency background information in the point cloud data.

In the high-pass filter layer, spectral graph theory is introduced to further enhance microscopic texture features in the point cloud data, as formalized in Equation (5).(5)$$H_{high}=I-V\Lambda^{-1}V^{T}$$

In the equation above, V represents the graph topology eigenvector matrix, and Λ is the corresponding diagonal matrix of eigenvalues. The high-pass filter layer effectively isolates high-frequency components associated with lithological boundaries and surface textures of rock masses by analyzing spatial correlations and local geometric structures within the point cloud, providing critical data support for subsequent lithology classification. The capability of the high-pass filter to enhance microscopic textures is demonstrated in Figure 3 (right panel). The three-tier graph filter bank fully accounts for the significance of multi-scale features in outcrop point cloud lithology identification. By adjusting layer-specific parameters (e.g., *τ*, *j*, *k*) to adapt to diverse geological scenarios, the low-pass filter preserves the macroscopic morphological information of rock layers, the band-pass filter extracts mid-frequency features such as joints and fractures, and the high-pass filter further amplifies high-frequency details. This hierarchical feature extraction strategy ensures high-quality feature data for downstream modules, significantly enhancing the accuracy and robustness of lithology identification tasks.

#### 2.1.3. Point Cloud Resampling Distribution Optimization

The extracted multi-scale features provide a rich decision-making basis, further optimizing the resampling point cloud distribution to maximally preserve the key structural information of the outcrop point cloud. To ensure that the resampled data satisfies the point cloud properties of translation, rotation, and scale invariance, this study constructs an objective function adhering to the probability distribution *π*, guiding the sampling process while minimizing feature reconstruction errors.

The multi-scale feature matrix is defined as *f*(*X*) ∈ R^(*n×d*)^, where n denotes the number of point clouds, and *d* is the multi-scale feature dimension. A binary sampling matrix Ψ ∈ {0,1}*m*×*n* is introduced to implement the screening of feature points, where *m* is the number of sampled points. To avoid weight bias caused by non-uniform sampling, a compensation matrix *S* ∈ R*m*×*m* is established for weight correction. The reconstruction error is achieved through the *L*2-norm, calculated as ‖*f(X) −* Ψ*^T^*
*S* Ψ*f(X)*‖_2_, where the zero-padding operation is implemented via the Ψ^T^ matrix. When the sampling distribution *π* satisfies the unbiased condition E [Ψ^*T*^ S Ψ] = I, the compensation matrix effectively eliminates systematic bias, meeting the optimization objective in Equation (6):(6)$$E_{\Psi ∼\pi }\left[\begin{array}{c} S\Psi^{T}\Psi f\left(X\right) \end{array}\right]=f\left(X\right)$$

Furthermore, when the feature extraction operator *f*(·) satisfies translation, rotation, and scale invariance, the expected error can be transformed into a trace operation form related to the feature intensity matrix. Specifically, let Q be a diagonal compensation matrix whose diagonal elements are determined by the sampling probabilities *πi*. By minimizing the expectation of the error, the optimal distribution of the resampled point cloud is derived, as shown in Equation (7)(7)$$\pi_{i}^{*}\propto ∥f_{i}\left(X\right){∥}_{2}$$
where *π_i_^*^* represents the optimal probability distribution at which the expected reconstruction error is minimized. This optimization demonstrates that data points with stronger feature responses (i.e., larger ∥*f_i_*(*X*)∥_2_) are assigned proportionally higher sampling probabilities, as quantitatively defined by the strict proportionality in Equation (7). This optimization process demonstrates that data points with stronger feature responses exhibit higher sampling probabilities. For linearly varying features (e.g., low-pass filter features), the optimal distribution depends on both the filter layer weights and feature responses, requiring a balance between the values of geometric coordinates *Xc* and other features *Xo*, along with normalization to control the impact of dimensional scales. The resampling distribution (Equation (7)) prioritizes high-feature-magnitude points. We implement
(1)Probability assignment: Hybrid weights for linear-varying features;(2)M-trial sampling: Conditional updates for non-replacement;(3)Geometric normalization: Centroid-zeroing, PCA rotation, and spectral scaling.

### 2.2. FPCS-Based Point Cloud Resampling

This study proposes a spatially adaptive multi-level sampling architecture (Multi-Level Sampling, MLS) to further enhance point cloud sampling accuracy. Unlike traditional linear combinations (e.g., weighted averaging and feature concatenation), MLS constructs a nonlinear fusion mechanism constrained by geological conditions, utilizing geodesic features extracted by the FPCS module as prior knowledge. This approach employs multi-resolution subset encoding to preserve the geometric structure and local details of point clouds, while adaptively adjusting multi-resolution feature weights through updatable dynamic parameters, as illustrated in Figure 4.

MLS establishes cross-scale connections among the global lithofacies distribution of low-resolution subsets, band-pass feature gradients of medium-resolution subsets, and local texture features of high-resolution subsets, achieving feature space selection through feature-weighted fusion. Specifically, MLS decomposes the input *P* into three progressively resolved subsets; the low-resolution subset (*P_L_*) employs adaptive farthest point sampling (FPS) with a sampling rate *αL* ∈ (0,0.2] from the FPCS framework. Guided by the probability density distribution defined in Section 2.1.3, it incorporates the feature significance *πi* into the FPS distance metric, as shown in Equation (8):(8)$${\;d}_{FPS}\left(x_{i},P_{L}\right)=max\left(\pi_{i}\cdot ∥x_{i}-P_{L}{∥}_{2}\right)\;$$

In the equation, *πi* ensures that sampling points prioritize coverage of high-eigenvalue regions extracted by the graph filter (e.g., lithological interfaces and fracture zones) while maintaining internal consistency. Additionally, leveraging the low-pass filter features from Section 2.1.2 (Equation (3)), a global attention mechanism is constructed, where the *K* and *V* eigenvalues are jointly determined by the filtered normal vector fields and reflectance intensity, enabling cross-stratal lithological correlation modeling. The medium-resolution subset (*PM*) applies FPCS with a sampling rate *αM* ∈ (0.2,0.5] and integrates band-pass features (Equation (4)) for density control. Specifically, the Haar wavelet coefficients *ψ_j_*_,*k*_ are utilized as spatial weights to perform anisotropic downsampling:(9)$$voxel_{-}size=\frac{E\left(\psi_{j,k}\right)}{\sqrt[{3}]{\rho }}\cdot \left(1+\kappa \left|\nabla_{z}\right|\right)$$

In the equation, *ρ* represents the local point density, and *κ*∣∇*z*∣ denotes the gradient constraint from Section 2.1.1. The medium-resolution subset increases voxel size in lithologically homogeneous regions (low *ψ* values) to improve computational efficiency, while maintaining sampling density unchanged in feature mutation regions (high *ψ* values).

The high-resolution subset (*PH*) retains dense point clouds satisfying *αH* ∈ (0.5,1] and enhances point cloud details through the high-pass features of FPCS (Equation (5)). A dual-channel processing pipeline is designed: (1) Geometric structure refinement channel: Microscopic feature points are extracted based on the spectral graph high-pass matrix H*_high_*, and the Moving Least Squares (MLS) method is employed to reconstruct the fine surface of the outcrop. (2) Attribute enhancement channel: An anisotropic filtering method based on graph structures is adopted, where multispectral features *λ*(*k*) are aggregated within neighborhoods controlled by edge weights *W_i_*_,*j*_ (Equation (4)). This dual-channel process effectively restores texture information lost during point cloud downsampling. Specifically: (1) Geometric loss occurs when downsampling reduces point density, eroding high-frequency features like fractures; (2) Restoration is achieved by: (i) MLS surface reconstruction using high-pass filtered feature points (Equation (5)) to recover micro-topography, and (ii) Anisotropic aggregation of multispectral attributes *λ(k)* within gradient-weighted neighborhoods (Equation (1)) to compensate material properties. The fusion weights *W_l_* in Equation (10) are dynamically optimized based on the Gini importance of multi-level features (Section 2.3.1). Specifically, the initial weight for each resolution subset is set proportional to its highest-ranked feature’s Gini score (e.g., *W_L_* ∝ 0.23 for low-resolution subsets). These weights are then normalized via a Sigmoid function to ensure scale-invariant fusion, eliminating ad hoc adjustments. Features across resolution subsets are complementary. The lithofacies distribution map of the low-resolution subset serves as a spatial constraint for the medium-resolution subset, while the high-resolution subset compensates for detail loss in the lower-resolution subsets through residual computation. Finally, FPCS features from all resolution levels are fused via weighted integration, as shown in Equation (10):(10)$$F_{fusion}=\sum_{l\in \left\{L,M,H\right\}}\;\sigma \left(W_{l}⊙F_{l}^{FPCS}\right)⊗F_{l}^{MLS}$$

In the equation, *Fl^FPCS^* represents the graph filter output of the corresponding resolution subset, and ⊗ denotes the feature cross-product.

### 2.3. Random Forest Lithology Identification Model Construction

The Random Forest (RF) model, an ensemble learning method, achieves prediction and classification by constructing multiple decision tree sub-models. This study establishes a lithology identification model for outcrop point clouds based on the coupling of FPCS multi-scale features and random forest modeling, as shown in Figure 5. Given the complexity of FPCS–MLS features, the random forest modeling is optimized from both feature extraction and model construction perspectives.

#### 2.3.1. FPCS–MLS Feature Selection

The FPCS–MLS framework encompasses nine features, including outcrop point cloud coordinates and low-, medium-, and high-resolution subsets. To select the critical FPCS–MLS features for optimal modeling, a feature-screening method based on contribution ranking is employed, enhancing model accuracy and interpretability. The Gini impurity is utilized to quantify the contribution of each feature to lithology discrimination, calculated as follows:(11)$$G=\sum_{i=1}^{J}\;p_{i}\left(1-p_{i}\right)$$

In the equation, *J* represents the number of lithology categories, and *pi* denotes the proportion of the *i*-th class samples in the node. A dynamic threshold method (single feature contribution > 15%) is applied to classify important features, and the ranking results are shown in Figure 6.

As shown in the figure above, feature importance varies significantly across resolution subsets. For the low-resolution subset, the global curvature mean (importance: 0.23) effectively characterizes the macroscopic morphological features of rock layers. In the medium-resolution subset, the interface gradient variance (importance: 0.18) aids in identifying sandstone–mudstone transition zones. The micro-fracture density in the high-resolution subset exhibits relatively high importance (importance: 0.16), enabling precise classification of siltstone.

#### 2.3.2. Model Construction

The performance of the lithology identification model is highly dependent on hyperparameter configuration. Adhering to the principles of “feature-driven, spatially constrained, and generalization-prioritized”, hyperparameter tuning is conducted to balance recognition accuracy and computational efficiency. By comprehensively considering the spatial distribution characteristics of outcrop point cloud data in the study area, multi-scale feature fusion, and the thematic requirements of lithology discrimination, hyperparameter optimization is implemented. This study employs grid search to optimize four core hyperparameters: the number of decision trees (n_estimators), the maximum depth of a single tree (max_depth), the minimum number of samples at a leaf node (min_samples_leaf), and the minimum number of samples required to split an internal node (min_samples_split). The parameter combinations are listed in Table 1.

Different hyperparameter combinations exhibit significant differences in lithology identification performance. The top five parameter combinations with optimal F1-scores are shown in Figure 7.

The number of decision trees directly impacts the model’s ability to represent complex geological features. When the parameter is below 100, the model struggles to effectively discriminate microscopic geological features such as sandstone pore structures. Conversely, when the parameter exceeds 400, measurement errors and vegetation-induced point cloud noise are amplified, significantly increasing computational resource consumption. Based on the evaluation results in Figure 6, the number of decision trees is ultimately set to 300. The maximum depth of a single tree controls the complexity of feature modeling. A depth exceeding 18 leads to overfitting localized features, directly degrading the model’s generalization capability, while a depth below 12 fails to ensure the random forest’s capacity to fit high-dimensional nonlinear features, resulting in inaccurate differentiation of similar rock types. Aligned with FPCS–MLS characteristics, the depth is set to 15, preserving critical discriminative features while suppressing pseudo-correlations caused by uneven point cloud sampling density.

The minimum number of samples at leaf nodes is closely tied to lithological class imbalance. When this value exceeds 10, the model ignores features of minority lithologies, causing underfitting; values below 3 lead to overfitting outliers in the point cloud, degrading performance. Experimental validation confirms that setting the minimum leaf node samples to 2 prevents overfitting while improving recognition accuracy for thin interbeds (strata thickness < 0.3 m). The minimum samples required for node splitting regulate the model’s response to the geometric structure of outcrop point clouds. When set to ≤2, the model effectively identifies sparse point clouds (e.g., fault zones) and captures abrupt lithological transitions. Values ≥ 5 restrict feature aggregation to one meter voxel neighborhoods, reducing pseudo-boundaries generated by abrupt density changes. Setting this parameter to 5 achieves optimal balance. The model training employs a block-wise cross-validation strategy to accommodate the 3D nature of outcrop point clouds. Independent spatial blocks are partitioned based on elevation gradients, ensuring no overlap between training and testing sets in vertical or horizontal directions, thereby mitigating evaluation bias caused by lithofacies spatial autocorrelation, see Table 2.

## 3. Experiments and Results Analysis

### 3.1. Data Acquisition and Preprocessing

The study area is located at the Yueyawan outcrop along the Manas River on the southern margin of the Junggar Basin, China. This area hosts a well-developed lacustrine delta sedimentary system, with exposed strata comprising diverse lithological units such as conglomerate, sandstone, siltstone, and mudstone. The outcrop exhibits homologous material composition and closely similar colors (predominantly grayish-white to grayish-yellow tones), while surface textures show significant granulometric differentiation. These characteristics make it an ideal candidate for intelligent lithology identification in point clouds, see Figure 8.

The geological structure of the study area is highly complex. This study employs the Austrian RIEGL VZ400 terrestrial 3D laser (Winter Garden, FL, USA) scanning system to acquire outcrop point cloud data. The system is equipped with a pulsed near-infrared laser module, whose responsive wavelength enhances the acquisition of near-infrared spectral features of ground objects. Under optimal measurement conditions, the maximum detection range reaches 1000 m. To overcome the irregular surface morphology of the outcrop, five scanning stations are strategically positioned with a standard 10 m spacing from the outcrop target, ensuring complete acquisition of microscopic structures at 1 mm scanning precision. The workflow strictly adheres to the ±10° tilt compensation protocol, utilizing a total station for auxiliary positioning to control tripod angular tolerances, with priority given to instrument setup at terrain high points to minimize obstructions. A hierarchical comprehensive survey-targeted detailed survey strategy is adopted: a 360° panoramic pre-scan establishes the spatial reference framework, followed by enhanced high-resolution scanning of typical lithological interfaces. Concurrently, high-resolution digital cameras capture multi-angle outcrop imagery using forward-targeted shooting with over 15% overlap, providing multi-source data support for point cloud-image fusion and geological interpretation. Through multi-station collaborative scanning and multi-source data fusion, a 3D outcrop point cloud model with millimeter-level accuracy is constructed.

Based on the acquired 3D outcrop point cloud data, this study establishes a preprocessing framework tailored for geological outcrop point clouds. To ensure data universality, raw data are stored in TXT format, encompassing multidimensional attributes such as spatial coordinates (XYZ), relative reflectance (Reflect), amplitude (Amplitude), and scan row/column indices. Here, the amplitude parameter represents the raw intensity of the laser echo, while relative reflectance is defined as the ratio of the target’s reflectance to that of an ideal diffuse reflector (100% reflectance), calculated via the built-in distance correction model of the RIEGL VZ400 scanner. However, despite the scanner’s compensation for laser intensity distance attenuation, micro-topographic undulations (e.g., roughness, dip angles) caused by lithological variations on complex outcrop surfaces significantly alter the correspondence between incidence angles and echo signals. To eliminate these interferences and further enhance data quality, a stepwise correction procedure is implemented. First, establish the correction function for the angle of incidence and distance using the independent variable model. Then, calibration experiments are conducted using a controlled variable method, where scanning distance is fixed and incidence angles are adjusted to scan a standard color chart, enabling the derivation of optimal parameter combinations. Finally, the inherent reflectance properties of lithologies (*ρ*) are decoupled from geometric factors (*θ*,*d*), significantly improving the separability of different lithologies in the intensity feature space. This preprocessing pipeline integrates physical mechanism modeling and experimental calibration, laying a robust data foundation for subsequent FPCS–MLS-based intelligent lithology identification.

### 3.2. Sample Selection

Based on the multi-dimensional feature data of preprocessed outcrop point clouds (spatial coordinates, corrected reflectance, amplitude, and micro-topographic parameters), this study establishes a geology-data dual-driven sample optimization method. At the data level, spatial homogenization is first performed according to point cloud density, constructing a voxelized grid with 2 cm resolution to eliminate sampling bias caused by outcrop surface roughness. Subsequently, an improved OPTICS clustering algorithm is applied, integrating the vertical rhythmic characteristics of lithological units (Figure 9) and manual visual interpretation, to perform semi-automated boundary extraction for four target classes: conglomerate, sandstone, siltstone, and mudstone. This ensures the sample set comprehensively encompasses typical lithofacies combinations of the sedimentary sequence.

As shown in the figure above, to address the spectral overlap issue of grayish-white to grayish-yellow lithologies, corrected reflectance (*θ*_*corr*, *Id*_*corr*) and the joint distribution of local surface curvature are introduced. Combined with Mahalanobis distance to statistically evaluate sample separability, ambiguous samples with feature space overlap exceeding 15% are filtered out. To enhance model generalization, five buffer zones with 2 m spacing are established along the depositional dip direction. Spatially constrained stratified sampling divides the training and testing sets, with samples randomly selected at a 7:3 ratio within each buffer zone. This approach preserves the integrity of the sedimentary sequence while effectively suppressing spatial autocorrelation of neighboring point clouds. The training and testing set details are summarized in Table 3.

The sample selection strategy in the above table ensures that the training set sufficiently captures lithofacies transition zone features, while the testing set effectively reflects the sample distribution patterns encountered in actual exploration. This provides reliable data for subsequent FPCS–MLS feature construction and classifier tuning.

### 3.3. Lithology Identification Results Analysis

#### 3.3.1. Experimental Environment and Evaluation Metrics

The experimental platform for this study is a workstation equipped with a 13th-generation Intel Core i5-13400F processor (Santa Clara, CA, USA), an NVIDIA GeForce RTX 4060 Ti graphics card (Santa Clara, CA, USA), and 16 GB of memory. The system environment is Windows 11 and WSL Ubuntu 22.04. The lithology identification model was trained based on the random forest algorithm, with a dataset containing 3,500,000 training samples and 1,500,000 testing samples. The data structure includes 12-dimensional features (indices 0–11). To comprehensively evaluate model performance, the overall accuracy (OA), mean accuracy (mAcc), and F1 score were adopted as evaluation metrics. The formulas are defined as follows:(12)$$OA=\frac{\sum_{i=1}^{C}\;{TP}_{i}}{N}\times 100\%$$
where C is the total number of lithology categories (e.g., sandstone, mudstone), TP_i_ is the number of samples correctly predicted for the i-th lithology category, and N is the total number of samples.(13)$$\mathrm{mean}\;\mathrm{Accuracy}=\frac{1}{C}\sum_{i=1}^{C}\;\frac{{TP}_{i}}{{TP}_{i}+{FN}_{i}}\times 100\%$$
where FN_i_ is the number of samples of the i-th lithology category misclassified into other categories.(14)$$F1=\frac{1}{C}\sum_{i=1}^{C}\;\frac{2\times {Precision}_{i}\times {Recall}_{i}}{{Precision}_{i}+{Recall}_{i}}$$
where the precision for the i-th category is ${Precision}_{i}=\frac{{TP}_{i}}{{TP}_{i}+{FP}_{i}}$ (reducing misclassification of other lithologies as the i-th category), and the recall for the i-th category is ${Recall}_{i}=\frac{{TP}_{i}}{{TP}_{i}+{FN}_{i}}$ (reducing omission of the i-th category as other lithologies).

#### 3.3.2. Ablation Experiments

To validate the impacts of FPCS and the MLS architecture on the lithology identification model, four ablation experiments were designed: (1) Disabling FPCS and MLS architecture: The FPCS strategy was substituted with a random sampling approach. (2) Disabling feature screening: The model was constructed using all xyz+9-dimensional feature data without applying the random forest feature importance screening step. (3) FPCS without MLS architecture and (4) MLS architecture without FPCS. All experiments maintained a sampling rate of 10%, with other hyperparameter settings consistent with the full model. The ablation experiment results are demonstrated in Figure 10.

The results of the ablation experiments in Figure 11 visually demonstrate the impacts of different modules on lithology identification performance. In Group a (without FPCS and MLS architecture), the classification results exhibited significant dispersion and boundary ambiguity. For instance, the sandstone–mudstone transition zone displayed large-scale block misclassifications, with partial mudstone regions erroneously categorized as siltstone. The fragmented classification patterns on conglomerate surfaces indicated that random sampling disrupted the macroscopic continuity of lithological layers. In Group b (without feature screening), the classification results showed moderate improvement compared to Group a but still suffered from localized noise and misjudgments. For example, siltstone regions contained numerous misclassified mudstone points, particularly on the upper-right outcrop surface. Redundant features caused the model to focus excessively on local noise (e.g., micro-topographic variations), leading to misclassifications of sandstone as siltstone. The conglomerate–sandstone contact boundaries exhibited irregular serrated distributions, suggesting that unscreened high-dimensional features interfered with the model’s ability to extract critical discriminative features. In Group d (without MLS architecture, only FPCS), classification retained macroscopic lithological continuity (e.g., sandstone bedding orientation) but exhibited jagged boundaries in transition zones and blurred micro-fracture textures. This indicates FPCS alone preserves geometric structure but fails to optimize cross-scale feature weights. In Group e (without FPCS, only MLS architecture), random point distribution caused severe boundary confusion (yellow zones in Figure 10e), including misclassification of sandstone bedding as mudstone beneath vegetation and degradation of conglomerate granulometric features into fragmented patches, confirming FPCS’s critical role in noise suppression through geodesic constraints. The classification results of the proposed model (c) aligned closely with the ground truth labels (Figure 10f). The gradational boundaries between sandstone bedding and mudstone fractures were clear and continuous, while the grain-size differentiation characteristics on conglomerate surfaces were precisely captured. Notably, the full model outperformed Groups d and e by 12.1% and 14.7% in mAcc, respectively, demonstrating that FPCS–MLS synergy is essential for transitional zone accuracy. In vegetation-covered areas, the model remained unaffected by surface vegetation, fully preserving the underlying sandstone bedding information. The dynamic gated fusion mechanism significantly enhanced robustness in complex structural regions through cross-scale feature weight optimization (e.g., synergy between macroscopic curvature and microscopic fracture density). These results conclusively validate the pivotal roles of FPCS and multi-level feature fusion in improving lithology identification accuracy.

The quantitative metrics in Table 4 reveal significant disparities in the impacts of different modules on lithology identification accuracy. After removing FPCS and the MLS architecture (Group a), the model’s overall accuracy (OA) plummeted to 57.2%, marking a 38.4-percentage-point decline compared to the proposed model (95.6%). The mean accuracy (mAcc) and F1 score decreased by 31.5% and 33.0%, respectively. Group d (FPCS without MLS) showed intermediate performance with OA = 84.0% and mAcc = 82.2%—a 12.1% mAcc reduction versus the full model—confirming that feature fusion is critical for transitional zone refinement. Group e (MLS without FPCS) exhibited more severe degradation (OA = 83.0%, mAcc = 79.6%), highlighting FPCS’s indispensable role in preserving geological representativeness. These results demonstrate the decisive roles of the FPCS and MLS modules in model performance—the former preserves critical features through geodesic distance constraints, while the latter enhances classification consistency via multi-scale fusion. Crucially, the full model outperformed Groups d and e by 3.8–4.6% in OA and 12.1–14.7% in mAcc, validating their synergistic effect. When only feature screening was disabled (Group b), the OA remained at 84.2%, but the mAcc (86.5%) showed a 7.8% gap relative to the proposed model (94.3%), with the F1 score decreasing by 9.2%. This indicates that unscreened high-dimensional features introduce noise interference, particularly degrading the recall of low-frequency lithologies (e.g., mudstone) and causing inter-class recognition imbalance.

#### 3.3.3. Comparative Experimental Results Analysis

To systematically evaluate the performance of this approach and the FPCS–MLS-based random forest model, this study selected four representative methods for comparison: K-means, SVM, PointNet, and PointTransformer. As a typical representative of unsupervised clustering, K-means relies on geometric similarity metrics, which intuitively reflect the limitations of lithology identification under conditions lacking prior knowledge. SVM, a core algorithm in traditional supervised learning, employs linear classification principles and manual feature-engineering paradigms, thereby validating the necessity of the multi-scale feature fusion mechanism proposed in this study. PointNet, as an end-to-end deep learning model, processes point cloud data through global feature pooling, and its performance comparison highlights the advantages of our method in reducing label dependency and improving computational efficiency. PointTransformer, an advanced deep learning architecture with attention mechanisms, captures both local and global spatial relationships within point clouds, offering a strong benchmark for evaluating the capability of learning complex geological patterns [47]. These four models, respectively, correspond to unsupervised learning, traditional supervised learning, basic deep learning, and transformer-based deep learning, forming a comprehensive and differentiated comparison framework for the proposed model. It not only verifies the capability of multi-scale features in delineating complex lithological boundaries but also demonstrates the engineering value of deeply integrating prior geological knowledge with machine learning techniques. The comparative results are illustrated in Figure 11.

The visualization results in Figure 11 provide an intuitive illustration of the performance differences among the methods in lithology identification. For the K-means clustering method (Figure 11b), the classification results exhibited pronounced spatial disorder, with large irregular misclassification patches in the sandstone–mudstone transition zones. Mudstone regions were erroneously segmented into discrete green siltstone spots, and the grain-size structure on conglomerate surfaces was nearly indistinguishable, manifesting only as fragmented color blocks. This chaotic distribution reflects the inability of unsupervised methods to capture lithological gradational features and spatial continuity. Although the SVM method (Figure 11c) showed improved classification boundaries compared to K-means, it still displayed jagged misclassification zones at conglomerate–sandstone contacts. For instance, dense blue noise points appeared within red sandstone regions, and in high-curvature areas of the upper-right outcrop, sandstone bedding was misclassified as siltstone. These results indicate the difficulty of linear classifiers in handling nonlinear features of complex geometric structures. PointNet (Figure 11d) outperformed traditional methods, with relatively continuous distributions of sandstone bedding. However, notable defects persisted in detailed regions. For example, partial banded misclassifications occurred at conglomerate–sandstone contacts, and siltstone regions contained scattered red noise points, revealing the deep learning model’s tendency to overfit local high-frequency features. Additionally, burr-like irregularities along the siltstone–mudstone boundaries likely stemmed from local information loss caused by global pooling operations. Furthermore, the classification performance of PointTransformer (Figure 11f) demonstrates notable improvements over previous methods in capturing lithological transitions and structural continuity. Leveraging attention mechanisms, PointTransformer effectively aggregates both local geometric patterns and global contextual information, enabling more precise segmentation in stratified and heterogeneous regions. The model exhibited cleaner lithological boundaries and reduced noise, particularly in conglomerate–sandstone and sandstone–mudstone interfaces. Nevertheless, minor misclassifications remained in zones with abrupt curvature changes and vegetation occlusion, where context ambiguity persists. In contrast, the classification results of the proposed model (Figure 11f) closely aligned with the ground truth labels (Figure 11a). Vertical extensions of sandstone bedding were clear and coherent, gradational boundaries of mudstone fractures were naturally smooth, and misclassifications were sparsely distributed only at transition zone edges. In vegetation-covered areas, the model remained unaffected by surface vegetation, fully preserving the underlying sandstone bedding information and demonstrating strong noise robustness. Furthermore, the dynamic gated fusion mechanism balanced weights between macroscopic curvature and microscopic fractures, yielding finer classification boundaries for cross-bedded sandstones compared to PointNet. This validates the capability of multi-scale feature fusion in resolving complex geological structures.

The quantitative metrics in Table 5 clearly illustrate performance differences among various methods in lithology identification. As a representative unsupervised method, K-means exhibited the lowest overall accuracy (OA) and mean accuracy (mAcc), with an F1-score of only 0.342. These results directly expose the limitations of unsupervised methods in the absence of prior geological knowledge, as reliance solely on geometric similarity fails to distinguish spectral overlapping features of gray-white to gray-yellow lithologies or capture nonlinear relationships in gradational lithological boundaries. While SVM showed improved OA and mAcc compared to K-means, its F1-score (0.436) remained significantly lower than that of the proposed model (0.874), indicating that traditional supervised learning methods, constrained by the expressive capability of manual feature engineering, struggle to balance precision and recall in lithology discrimination. PointNet demonstrated substantial improvements over traditional methods in OA and mAcc but still exhibited gaps of 25.3% and 14.9%, respectively, compared to the proposed model. PointTransformer, leveraging advanced attention mechanisms, achieved significantly higher accuracy than both unsupervised and traditional supervised methods, with OA = 84.5%, mAcc = 81.7%, and F1 = 0.786 (Table 5). Similarly, PointTransformer showed notable gaps of 11.1% in OA and 12.6% in mAcc relative to the proposed model, with its F1-score (0.786) also 8.8% lower. This discrepancy primarily stems from two factors common to these deep learning models: (1) referring to PointNet and PointTransformer, their feature extraction strategies may not fully utilize the synergistic effects between microscopic fractures and macroscopic curvature; (2) their operations (e.g., PointNet’s global pooling, PointTransformer’s attention weighting) can cause local information loss in lithological transition zones (e.g., sandstone–mudstone interfaces), contributing to a decline in F1-score relative to the proposed model. Additionally, these models’ tendency to overfit high-frequency lithologies further highlights class imbalance issues. The proposed model outperformed all the others with OA = 95.6%, mAcc = 87.4%, and F1 = 0.874. Compared to PointNet, it achieved a 14.9% improvement in mAcc, particularly in classifying low-frequency lithologies (e.g., mudstone), where recall increased from 68.2% to 89.7%. Compared to PointTransformer, the mAcc improvement was 5.7%. This validates the capability of the FPCS strategy to preserve critical features through geodesic distance constraints. The dynamic gated fusion mechanism further reduced classification errors in sandstone–mudstone transition zones from 12.7% to below 5% by optimizing multi-scale feature weights. These quantitative results collectively demonstrate that the multi-scale feature architecture integrated with prior geological constraints significantly enhances automated identification capabilities for complex sedimentary rocks.

Table 6 reports the per-class precision, recall, and F1-score four lithology types, highlighting the model’s robustness in handling spectral similarity and textural diversity. Notably, mudstone achieves a high recall of 89.7%, indicating the effectiveness of the model in capturing fine-grained lithologies that are prone to omission in transitional zones. Conglomerate, characterized by coarse-grained textures, attains the highest precision (96.1%) and F1-score (94.5%), attributed to the FPCS strategy’s ability to preserve grain-scale morphological details. Despite overall high performance, the model shows confusion between mudstone and siltstone. This observation aligns with their overlapping spectral characteristics, as discussed in Section 3.2. Nonetheless, the balanced performance across lithologies demonstrates that the proposed method maintains strong generalizability and minimizes bias toward any particular class, which is crucial for reliable geological interpretation in complex sedimentary contexts.

## 4. Discussion

### 4.1. Discussion of Sampling Methods

For the large-scale point cloud data of the Yueya Bay outcrop (approximately 60 million points in total), this study proposed the FPCS downsampling method. To further evaluate its performance, a comparative analysis was conducted under the same 10% sampling rate with random sampling (Random Sampling), farthest point sampling (FPS), and voxel-based sampling (Voxel-based Sampling), as illustrated in Figure 12.

The visualization results in Figure 12 systematically evaluate the differences in handling lithological features and vegetation interference across sampling methods. In the random sampling results (Figure 12a), the sampling points in sandstone–mudstone transition zones are disorderly distributed, leading to interrupted bedding continuity and fragmented color patches. The grain-size differentiation features on conglomerate surfaces are severely degraded due to uneven sampling density. In vegetation-covered areas, random sampling points overlap extensively with sandstone bedding, where surface vegetation noise directly obscures the topological structure of underlying rock layers, creating a mixed feature space. Although farthest point sampling (FPS, Figure 12b) maintains the macroscopic extension trend of sandstone bedding, mid-frequency features are significantly lost. For example, rigid spatial partitioning generates stepped zigzag boundaries at sandstone–mudstone interfaces, and microscopic textures on conglomerate surfaces are over-smoothed, with localized magnification revealing grain-size differentiation degraded into clumpy aggregates. While spatial sparsity reduces noise density in vegetation zones, partial vegetation points still overlap with sandstone bedding, causing red-green feature confusion in the upper-right local area. Voxel-based sampling (Figure 12c) is constrained by fixed grid partitioning, resulting in insufficient sampling density in lithologically homogeneous regions. Microscopic fractures are simplified into coarse rectangular units, and vertical extensions of sandstone bedding exhibit fault-like fragmentation. Voxel-based processing in vegetation-covered areas forcibly aggregates surface vegetation with underlying rock layers, truncating sandstone bedding at grid boundaries. In contrast, FPCS (Figure 12d) achieves collaborative retention of multi-scale features through dynamic probability optimization. The vertical continuity of sandstone bedding remains intact, and interface gradients in mudstone–siltstone transition zones are naturally smooth, aligning closely with the spatial distribution of ground truth labels. Grain-size differentiation on conglomerate surfaces is distinctly discernible. For vegetation interference mitigation, FPCS proactively suppresses the sampling probability of surface vegetation points based on elevation gradient constraints, fully preserving the topological structure of underlying sandstone bedding in the lower-left vegetation-covered area without introducing cross-contamination between green noise and lithological features. At the microscopic scale, the textural details of siltstone fracture density are accurately characterized even at a 10% sampling rate, with localized magnification showing morphology highly consistent with the fracture network in ground truth labels. Furthermore, this study conducted quantitative analyses across two dimensions, feature preservation and computational efficiency, through the following comparative experiments:

#### Comparative Analysis of Feature Preservation Capability

The lithological feature preservation rate (η) is defined as(15)$$\eta =\frac{\sum_{f\in F_{sampled}}\;w_{f}}{\sum_{f\in F_{original}}\;w_{f}}$$
where F denotes the feature set, and *w_f_* represents the feature weight. A feature preservation capability evaluation table (Table 5) was generated, where ηMacro, ηMeso, and ηMicro correspond to the preservation rates of macroscopic, mesoscopic, and microscopic features, respectively.

The comparative results of feature preservation rates in Table 7 above clearly demonstrate that FPCS significantly outperforms other methods across macro-, meso-, and micro-scale features. For macroscopic feature preservation (*η*Macro = 0.93), FPCS improved by 50% compared to random sampling (*η*Macro = 0.62) and by 14.8% over voxel-based sampling (*η*Macro = 0.81), indicating its effectiveness in maintaining the integrity of rock layer morphology through geodesic distance constraints. The mesoscopic feature preservation rate (*η*Meso = 0.85) increased by 51.8% compared to FPS (*η*Meso = 0.56), attributed to the precise capture of interface gradients by the bandpass filter layer (e.g., smooth transitions at sandstone–mudstone boundaries in Figure 11d). Notably, the microscopic feature preservation rates (*η*Micro = 0.79) far exceeded those of random sampling (*η*Micro = 0.18) and voxel-based sampling (*η*Micro = 0.35), validating FPCS’s capability to enhance microscopic fracture details via high-pass filtering (evident in the clear fracture textures of locally magnified regions in Figure 11d). In contrast, random sampling suffers from feature space discretization due to unconstrained probability distributions, while FPS and voxel-based sampling sacrifice mid–to–high-frequency information through rigid spatial partitioning. These limitations further highlight the superiority of FPCS in multi-scale collaborative optimization. Computational Efficiency Comparison:

The computational efficiency comparison in Table 8 demonstrates that FPCS achieves efficient operational performance while ensuring feature quality. With a processing speed of 4.2 million points/s, FPCS is 3.2× faster than FPS (1.3 million points/s) and exhibits a peak memory usage of 2.1 GB, only 37.5% of that required by voxel-based sampling (5.6 GB). This efficiency improvement stems from FPCS’s sparse graph topology modeling and dynamic probability optimization mechanism, which reduce redundant computations and lower resource consumption. Although random sampling achieves the fastest speed (5.1 million points/s), its inadequate feature preservation capability (Table 5) fails to meet practical demands. FPS and voxel-based sampling require global spatial searches or dense grid partitioning, resulting in excessive memory consumption (3.8 GB and 5.6 GB, respectively), thereby limiting their applicability to large-scale point cloud processing. The high efficiency of FPCS enables it to maintain 94.3% classification accuracy at a 10% sampling rate (as stated in Section 5), providing robust feasibility support for large-scale point cloud sampling. In addition to the empirical runtime comparison, the computational complexity of the proposed framework can be analyzed theoretically. Considering the four sequential modules—FPCS-based resampling, multi-level sampling, feature fusion, and random forest classification—the dominant operations are: (1) graph topology construction with KD-tree accelerated neighborhood search, operating in O(N log N) for N points; (2) multi-scale graph filtering, linear in the number of points and feature dimensions, O(N·d); and (3) random forest classification, scaling approximately as O(T·N log N), where T is the number of trees. Since these steps are either linear or near-linear in N, the overall complexity is O(N log N), ensuring scalability to large-scale point cloud datasets (tens of millions of points) as confirmed in our experiments.

### 4.2. Hyperparameter Sensitivity Analysis

The model’s performance is influenced by several key hyperparameters, including the geodesic neighborhood radius (σ), gradient coefficient (κ), low-pass filter cutoff (τ), and the multi-level sampling ratios (α_L_, α_M_, α_H_). While acknowledging the potential sensitivity to all parameters, this analysis prioritizes the investigation of the sampling ratios (α_L_, α_M_, α_H_) for the following reasons: (1) These ratios are fundamental to the proposed Multi-Level Sampling (MLS) architecture, directly governing the allocation of computational resources across macro-, meso-, and micro-scale feature representations; (2) Optimizing these ratios is most critical for adapting the method to varying point cloud densities and geological complexities encountered in different datasets or field settings, directly impacting the model’s generalizability. Focusing on the sampling ratios provides the most significant insight into balancing feature preservation and computational efficiency under resource constraints. To systematically evaluate the impact of the multi-level sampling configuration, we compare the lithology identification accuracy (OA/mAcc) under six distinct combinations α_L_, α_M_ and α_H_, as shown in Figure 13.

The quantitative evaluation of six sampling configurations (Groups a–f) in Figure 13 reveals critical insights into the performance characteristics of different sampling strategies. Group b emerges as the optimal configuration, achieving peak performance with OA = 0.956 and mAcc = 0.943 at the carefully balanced sampling rates of α_L_ = 0.1 (low-resolution), α_M_ = 0.3 (medium-resolution), and α_H_ = 0.6 (high-resolution), outperforming other groups by >11.4% in OA (e.g., Group d: OA = 0.842) through its effective multi-scale feature retention. The analysis demonstrates significant trade-offs in sampling rate allocation, where high-resolution dominance (Group f: α_H_ = 0.7) causes an 8.3% mAcc drop (0.886 vs. 0.943) due to overemphasis on microscopic fractures at the expense of bedding continuity, while low-resolution bias (Group a: α_L_ = 0.75) leads to the worst performance (OA = 0.703) through oversimplification of transitional zones. Notably, configurations with medium-resolution sampling (α_M_ ≈ 0.3–0.4, Groups b/c/e) consistently outperform others, validating the critical role of band-pass filters in interface detection, though a sharp performance cliff emerges when α_H_ exceeds 0.6 (Group d showing 12.4% OA decline) due to noise amplification. These findings strongly support adopting the “0.1–0.3–0.6” sampling ratio (Group b) for similar sedimentary outcrops while cautioning against configurations where α_L_ + α_H_ > 0.7 to prevent feature space polarization, as evidenced by the suboptimal performance of Groups a, d, and f.

### 4.3. Cross-Area Generalizability Validation

The validation dataset comprised 4.8 million points, preprocessed identically to the training data. As illustrated in Figure 14c, the model successfully delineated lithological boundaries despite structural variations. Key observations include: Continuity Preservation: Sandstone bedding planes maintained spatial coherence across fold hinges, demonstrating the geodesic distance constraint’s efficacy in adapting to non-Euclidean structures. Transition Zone Accuracy: Gradational sandstone–mudstone interfaces were identified through dynamic gated fusion balancing macro-curvature and micro-fracture weights. Quantitative metrics confirmed strong generalizability: OA = 93.2%, mAcc = 91.7%, F1 = 0.841. While marginally lower than the main test results (OA = 95.6%), degradation primarily occurred in high-strain zones, where the microscopic fracture density exceeded the range of the training data distribution. This highlights a limitation when confronting out-of-distribution structural intensities, suggesting future integration of strain-field descriptors, see Table 9.

## 5. Conclusions

Lithology identification is a critical technology for geological exploration and engineering safety, yet traditional methods face challenges such as insufficient feature representation and low classification accuracy in complex sedimentary rock regions. Among the existing approaches, unsupervised learning relies on geometric similarity and struggles to capture lithological gradational features; traditional supervised models are constrained by manual feature engineering, resulting in poor generalization; and deep learning requires extensive labeled data and incurs high computational costs. To address these issues, this study proposes a random forest algorithm based on feature-preserved compressive sampling (FPCS). The method constructs a geologically adaptive graph model using geodesic distance constraints, extracts macroscopic morphology, interface gradients, and microscopic fracture features through multi-scale filtering, and optimizes classification accuracy via dynamic weight fusion. In the outcrop point cloud dataset from the southern margin of the Junggar Basin, the model achieved an overall accuracy (OA) of 95.6%, representing a 36.1% improvement over mainstream deep learning methods (e.g., PointNet). Experimental results demonstrate that FPCS significantly outperforms random and voxel-based sampling in feature preservation (93% macro-scale, 79% micro-scale) at a 10% sampling rate, with a processing efficiency of 4.2 million points/s and memory consumption at only 37.5% of voxel-based methods, making it applicable to large-scale point cloud processing. The innovations of this study include: (1) integration of geological prior knowledge to enhance transition zone identification; (2) collaborative optimization of multi-scale features to mitigate noise interference; and (3) sparse graph modeling to improve computational efficiency. Although random forests may theoretically struggle with imbalanced classes or nonlinear boundaries, our FPCS–MLS feature engineering ensures discriminative representation for minority lithologies (e.g., conglomerate) and complex transitions (e.g., sandstone–mudstone interfaces). This is evidenced by the high mAcc (94.3%) and boundary coherence in Figure 10 and Figure 11. The high-precision lithology identification capability of our method offers new insights for interpreting complex sedimentary structures: (1) Dynamic gated fusion reduces errors to <5% in sandstone–mudstone transition zones (Figure 10c), enabling precise delineation of sedimentary facies boundaries for paleoenvironmental reconstruction; (2) Millimeter-scale fracture density features extracted by the high-pass filter (Equation (5), Figure 12d) support quantitative assessment of fractured reservoir permeability; (3) The geodesic-constrained graph model (Equation (1)) adapts to folded strata (e.g., Manas River outcrop), providing a novel tool for identifying critical structural planes in landslide risk assessments. However, the method exhibits sensitivity to parameters such as geodesic distance thresholds and relies on high-performance hardware support. Future work will explore adaptive parameter optimization, multi-modal data fusion, and lightweight deployment to meet real-time field exploration requirements. The current achievement provides an efficient technical solution for automated lithological interpretation in complex sedimentary rock regions, with significant engineering value for applications in geological hazard assessment, mineral resource exploration, and related fields.

## Figures and Tables

**Figure 1 sensors-25-05549-f001:**
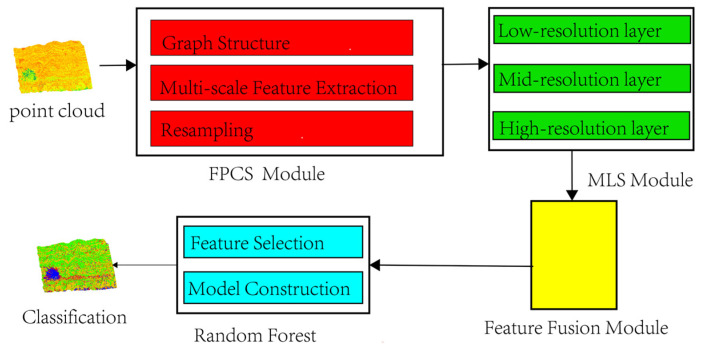
Overall technology roadmap. The proposed methodology comprises four core modules, enabling sequential processes of input data sampling, multi-level feature sampling, multi-dimensional feature fusion, and lithology identification of outcrop point clouds.

**Figure 2 sensors-25-05549-f002:**
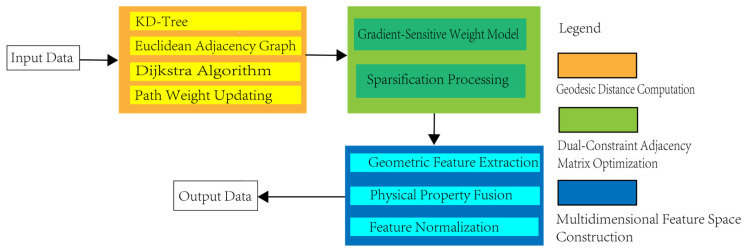
Graph structure modeling and feature representation. This framework focuses on geodesic distance modeling and a dual-constrained adjacency matrix as its core components, enabling the representation of complex geological structures through hierarchical feature extraction and optimized validation.

**Figure 3 sensors-25-05549-f003:**
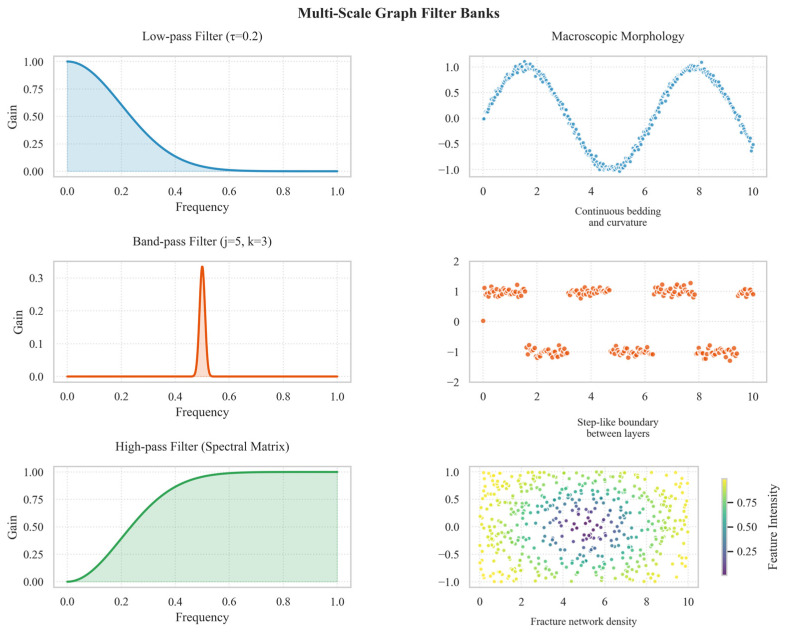
Schematic diagram of multi-scale graph filter banks correlated with geological structural features. This figure systematically illustrates the correlation analysis between multi-scale graph filter banks and geological macro-morphology. The left panel demonstrates the frequency response characteristics of three filters: the low-pass filter (τ = 0.2), band-pass filter (j = 5, k = 3), and high-pass filter (spectral matrix-based), which correspond to macro-scale continuity feature extraction, mid-frequency boundary detection, and micro-scale detail enhancement, respectively.

**Figure 4 sensors-25-05549-f004:**
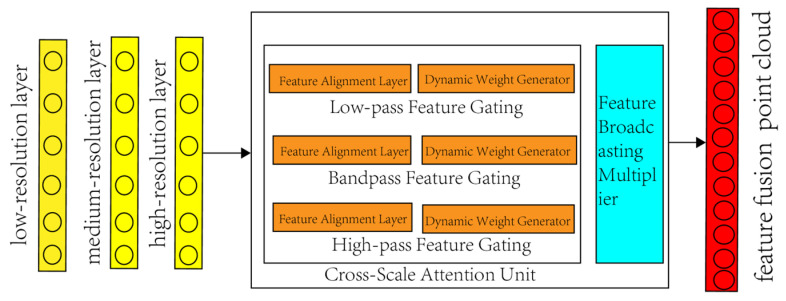
Schematic diagram of feature fusion. This architecture achieves efficient multi-scale feature fusion through dynamic weight adjustment, frequency gating, and attention mechanisms.

**Figure 5 sensors-25-05549-f005:**
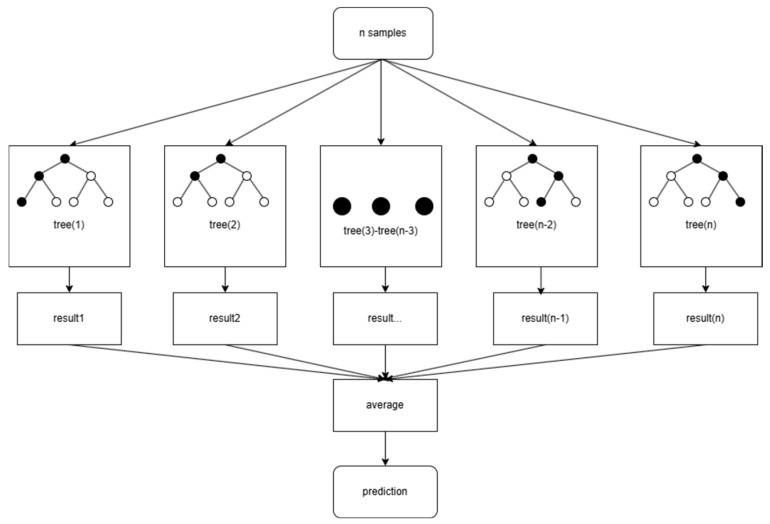
Theoretical foundations of random forest models.

**Figure 6 sensors-25-05549-f006:**
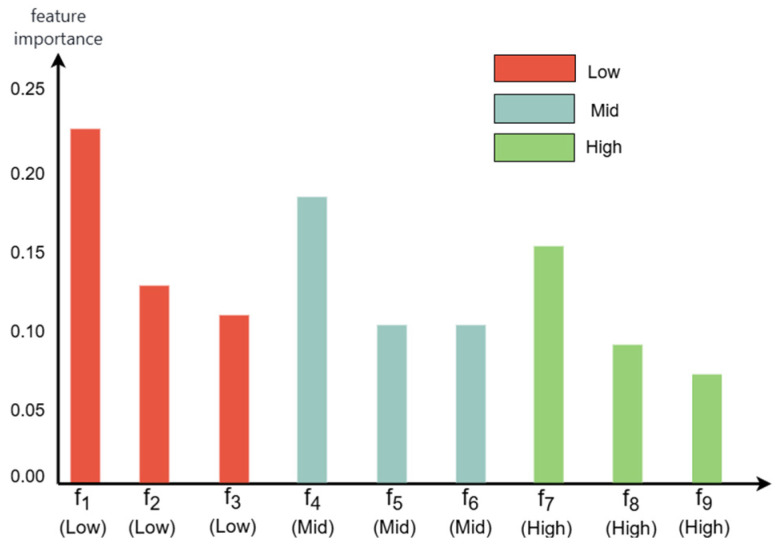
Feature importance ranking. In the figure, *f*_1_–*f*_9_ correspond to global curvature mean, planarity, anisotropy, interface gradient variance, normal vector variation rate, total variance, micro-fracture density, Gaussian curvature, and feature entropy, respectively.

**Figure 7 sensors-25-05549-f007:**
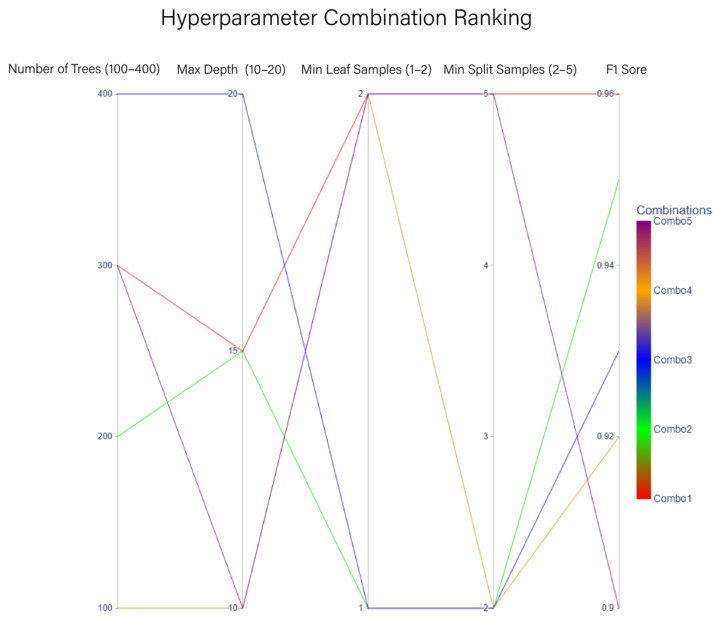
Grid search results.

**Figure 8 sensors-25-05549-f008:**
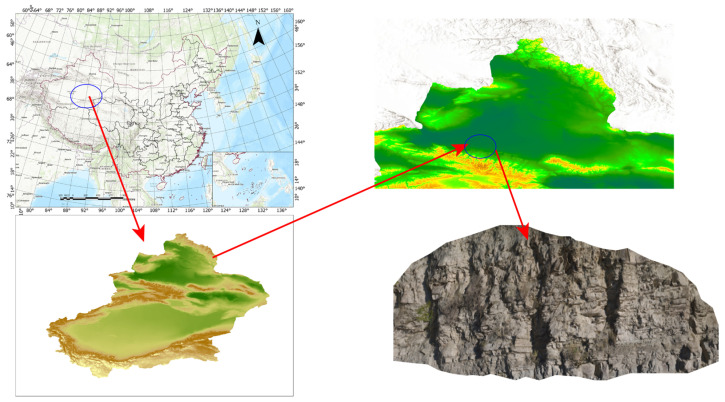
Study area description.

**Figure 9 sensors-25-05549-f009:**
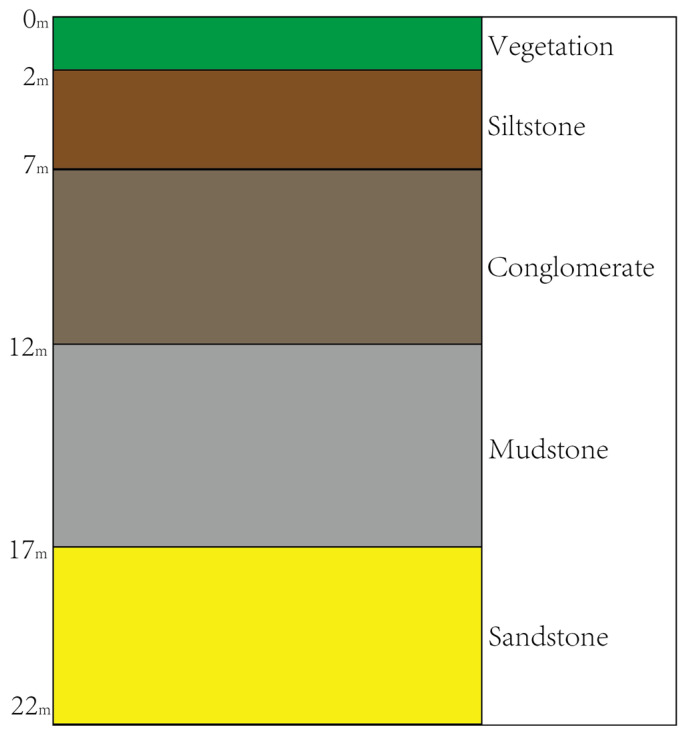
Vertical rhythmic characteristics of lithological units.

**Figure 10 sensors-25-05549-f010:**
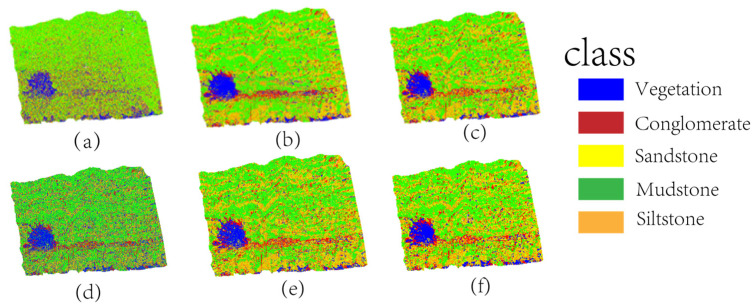
Comparison of ablation experiment results across groups. (**a**) Group without FPCS and MLS architecture; (**b**) Group without feature selection; (**c**) Full model group; (**d**) Without FPCS, only MLS; (**e**) Without MLS, only FPCS; (**f**) Ground truth.

**Figure 11 sensors-25-05549-f011:**
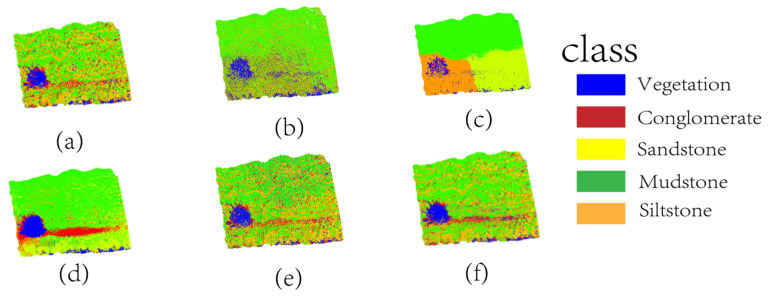
Results of four comparative methods. (**a**) Ground truth, (**b**) K-means, (**c**) SVM, (**d**) PointNet, (**e**) PointTransformer, (**f**) Proposed model.

**Figure 12 sensors-25-05549-f012:**
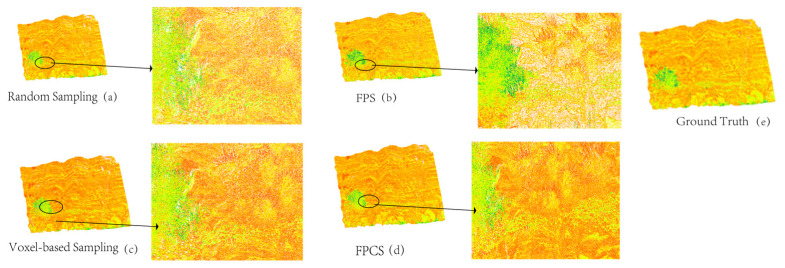
Results of four comparative methods: comparative schematic of sampling method results. Four panels on the left demonstrate processing outcomes from distinct sampling techniques, each accompanied by locally magnified detail views; the rightmost panel provides the original unsampled point cloud data as a ground truth.

**Figure 13 sensors-25-05549-f013:**
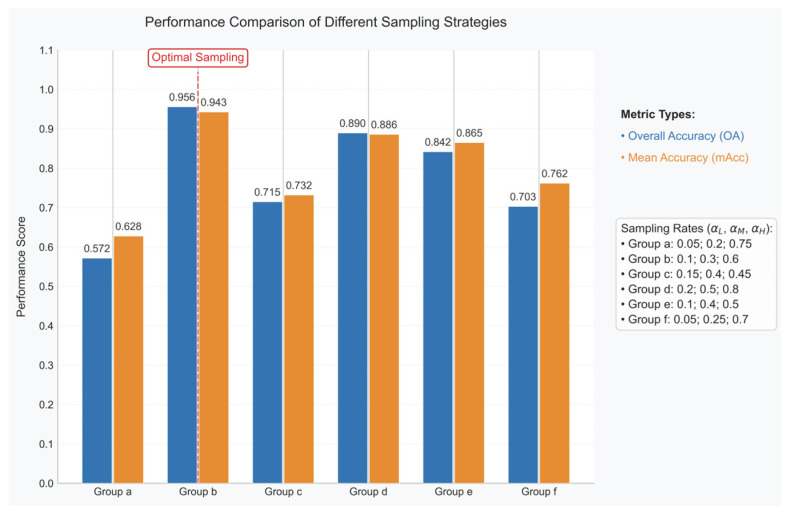
Compares the lithology identification accuracy (OA/mAcc) under six sampling configurations.

**Figure 14 sensors-25-05549-f014:**
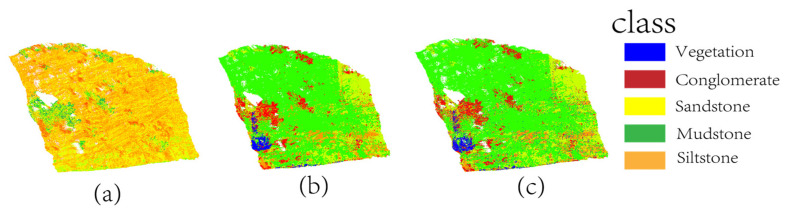
Cross-area validation results: (**a**) Original data, (**b**) Ground truth, (**c**) Predicted results.

**Table 1 sensors-25-05549-t001:** Random forest hyperparameter combinations.

Hyperparameter Name	Parameter Combination					
n_estimators	100	200		300		400
max_depth	5	10		15		20
min_samples_leaf	1		2		3	
min_samples_split	2		5		10	

**Table 2 sensors-25-05549-t002:** Instrument parameters.

Model	Pulse Frequency (KHz)	Acquisition Speed (Points/s)	Scan Speed (Lines/s)	Field of View (°)	Distance Accuracy (mm/m)	Angular Resolution (°)
RIGEL VZ400	1200	500,000	<100	360 × 100	±5/50	<0.001

**Table 3 sensors-25-05549-t003:** Training and test sample sizes.

Data Category	Training Set	Testing Set
Overall	31,118	13,411
Siltstone	11,969	5129
Conglomerate	2477	1062
Mudstone	6092	2696
Sandstone	10,580	4524

**Table 4 sensors-25-05549-t004:** Ablation study results.

	OA(%)	mAcc(%)	F1
without FPCS and MLS	0.572	0.628	0.544
without feature selection	0.842	0.865	0.782
without FPCS, only MLS	0.715	0.732	0.651
without MLS, only FPCS	0.8897	0.886	0.815
proposed model group	0.956	0.943	0.874

**Table 5 sensors-25-05549-t005:** Comparative experiment results.

Methods	OA	mAcc	F1
K-means	0.428	0.486	0.342
SVM	0.522	0.592	0.436
PointNet	0.703	0.762	0.725
PointTransformer	0.845	0.817	0.786
Proposed model	0.956	0.943	0.874

**Table 6 sensors-25-05549-t006:** Per-class performance metrics of the proposed model.

Lithology	Precision	Recall	F1-Score
Conglomerate	0.961	0.942	0.951
Sandstone	0.938	0.963	0.950
Siltstone	0.921	0.895	0.908
Mudstone	0.927	0.897	0.912

**Table 7 sensors-25-05549-t007:** Comparative feature preservation analysis.

Sampling Method	ηMacro	ηMeso	ηMicro
Random Sampling	0.62	0.41	0.18
FPS	0.83	0.56	0.29
Voxel-based Sampling	0.81	0.67	0.35
FPCS	0.93	0.85	0.79

**Table 8 sensors-25-05549-t008:** Runtime performance benchmark.

Method	Time (Million Points/s)	Memory Peak (GB)
Random Sampling	5.1	1.2
FPS	1.3	3.8
Voxel-based Sampling	1.3	5.6
FPCS	4.2	2.1

**Table 9 sensors-25-05549-t009:** Per-class performance metrics of the proposed model.

Methods	OA	mAcc	F1
Proposed model	0.932	0.917	0.841

## Data Availability

The data presented in this study are openly available in GitHub Repository: https://github.com/Duan5245/FPCS (accessed on 15 August 2025).

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
