# Peer review of "A Hierarchical Multi-Feature Point Cloud Lithology Identification Method Based on Feature-Preserved Compressive Sampling (FPCS)"

_sensors, 2025, doi:10.3390/s25175549_

Round 1

Reviewer 1 Report

Comments and Suggestions for Authors

This work presents a novel framework for lithology identification from 3D outcrop point clouds. The framework consists of three key feature extraction modules: Feature-Preserved Compressive Sampling (FPCS), Multi-Level Sampling (MLS), and hierarchical feature fusion. These are followed by a random forest-based lithology classification module. The method was tested on an outcrop dataset collected from the Junggar Basin, and results show a 36.1% improvement in overall accuracy and 20.5% in mean accuracy over the PointNet model. Furthermore, ablation study demonstrates the significant contributions of the FPCS and MLS modules to the overall performance.

  1. This paper proposes a novel framework for lithology identification. The innovations of this work include: the integration of geological prior knowledge to enhance transition zone identification; the joint optimization of multi-scale features to mitigate noise interference; the application of sparse graph modeling to improve computational efficiency.
  1. Instead of relying on a deep learning-based framework, this work adopts Feature-Preserved Compressive Sampling (FPCS) and Multi-Level Sampling (MLS) to extract hand-crafted multi-scale features, which are then combined with a random forest for classification. This setup eliminates heavy training requirements and is more suitable for geological point clouds, particularly in scenarios labeled data is scarce.
  2. The experimental evaluation is thorough, including both classification performance and system-level analysis. The proposed method is compared against K-means, SVM, and PointNet, consistently outperforming them in terms of overall and mean accuracy. Notably, it achieves a 36.1% improvement in overall accuracy and a 20.5% gain in mean accuracy compared with PointNet. Ablation studies further highlight the impact of the proposed modules, as well as a detailed comparison with other sampling methods such as FPS and voxel-based sampling. These results support both the effectiveness and practicality of the proposed framework.

Comments:

  1. The overall presentation of the paper would benefit from significant refinement. The methodology sections are difficult to follow, partly due to inconsistent variable notation (e.g., Wi,j and d_g(i,j) vs. ), which can cause confusion and reduce clarity in the mathematical formulation. In addition, several figures need improvement for better readability and consistency. For example, the font sizes in Figure 1 are uneven, and some labels are too small to read comfortably.  
  2. The comparative experiments are limited to relatively standard baselines such as K-means, SVM, and PointNet. The comparative analysis would be more compelling if it includes recent state-of-the-art models like PointTransformer [1].
  3. While the ablation study in table 4 test the impact of removing both FPCS and MLS together, the individual contributions of each module are not separately evaluated. Including such analysis would help clarify the role of each component in the overall performance.
  4. Figure 3 is currently not referenced or discussed in the main text.
  5. There are some formatting problems in this paper. For example, the caption of Figure 11 contains a mix of Chinese and English brackets. In addition, some tables lack the structured formatting typically expected in academic style and appear more informal in presentation. Improving the professionalism of figures and tables would enhance the overall quality of the paper.

Reference:

[1] Zhao, Hengshuang, et al. "Point transformer." Proceedings of the IEEE/CVF international conference on computer vision. 2021.

Reviewer 2 Report

Comments and Suggestions for Authors

This manuscript focuses on multi-feature hierarchical identification of lithological cloud points based on sampling. From an engineering, IT, and 3D modeling perspective, in order to make the manuscript more understandable even for those who are not specifically in the field, I suggest:

  • In line 23, it is mentioned that “they depend heavily on the intermediate results of feature engineering.” I recommend explaining the meaning of this sentence technically at this stage, referring to feature extraction methods. This is certainly a complex but necessary analysis that becomes a bottleneck for computational effort.
  • Line 95 mentions “points with higher eigenvalues have priority.” I recommend specifying technically and scientifically the relationship between the magnitude of the eigenvalues and the increase in priority, making considerations of mathematical proportionality and considering the mathematical relationship between the two entities involved.
  • Line 306 mentions “effectively restores lost texture information.” I recommend explaining what the restoration of information lost during downsampling consists of technically, and in particular how this information is lost and how it is recovered.
  • Line 344 mentions “For the low-resolution subgroup.” I recommend specifying more explicitly in the graph in Figure 6 the relationship between the features and the different resolutions. I recommend creating a more complete graph, with visible relationships between resolution and characteristics.
  • In addition, in order to improve the manuscript with useful insights, I suggest reading the following articles:

10.21203/rs.3.rs-7045774/v1

10.3390/s25051325

Reviewer 3 Report

Comments and Suggestions for Authors

In this manuscript, the authors propose a hierarchical multi-feature lithology identification method for 3D outcrop point clouds based on FPCS and a multi-level random forest classifier. The framework integrates graph signal processing with multi-scale feature extraction and dynamic fusion mechanisms to achieve high-accuracy classification in complex sedimentary environments. The work is well-positioned within the broader field of machine learning-based geological interpretation and is particularly relevant for applications requiring accurate lithology mapping from terrestrial laser scanning data.

However, several aspects of the manuscript warrant further attention and refinement:

  1. The manuscript is compact in content but tends to be overly wordy in some sections, particularly in the methodology. While the technical depth and rigor are commendable, certain descriptions, such as those related to graph modeling or resampling optimization, could be expressed more concisely and clearly. Streamlining the language would improve readability without compromising the scientific precision or completeness of the methods.
  2. The rationale behind the selected hyperparameters (e.g., specific values for τ in the low-pass filter, or sampling thresholds) could benefit from clearer justification. Currently, some values appear arbitrary, and their selection may significantly influence model performance.
  3. The choice of baseline models (K-means, SVM, and PointNet) is appropriate but limited. More recent graph-based or transformer-based models could have provided a stronger benchmark and contextualized the improvements more thoroughly.
  4. The ablation study is helpful, but further breakdown, such as isolating the effect of the geodesic distance modeling versus the gradient-sensitive weighting in FPCS, would add valuable insights into the contribution of each component.
  5. The manuscript would benefit from a stronger link to geological implications. While the classification accuracy is emphasized, more discussion on what geological insight this method enables would improve the practical relevance.
  6. The experiments are limited to one dataset. While the results are impressive, some discussion or evaluation on generalizability (e.g., across different lithological settings or sensor types) is missing and should be addressed to support broader claims.
  7. Although efficiency is mentioned and supported with empirical metrics, a more analytical or theoretical evaluation of the algorithm's computational complexity, especially in large-scale deployments, would strengthen the case for practical utility.
  8. The replacement of Euclidean distance with geodesic distance is a logical choice for non-planar topographies. However, the computational overhead associated with Dijkstra-based pathfinding on large-scale point clouds is significant. A discussion on algorithmic optimization or parallelization strategies would be beneficial.
  9. Although the use of graph filters (low-pass, band-pass, and high-pass) is well-justified in principle, the practical geological significance of each filter output could be elaborated upon.
  10. The proposed MLS architecture and dynamic gated fusion are central contributions. However, the fusion mechanism appears heuristic. Clarifying how fusion weights are optimized or learned beyond stating that they are "dynamically adjusted" would strengthen this section.
  11. The model's performance appears sensitive to several hyperparameters (e.g., sampling ratios, filter thresholds, and the gradient coefficient κ). A systematic sensitivity analysis would improve the reproducibility and help others adapt the method to different datasets or geological settings.
  12. While the use of a random forest offers interpretability and training efficiency, the paper does not discuss its limitations, especially in handling highly imbalanced classes or learning complex nonlinear decision boundaries.
  13. The model evaluation relies on OA, mAcc, and F1-score. However, confusion matrices or per-class precision/recall values would provide better insight into how well specific lithologies (e.g., fine-grained vs coarse-grained rocks) are being classified. In geological applications, class-specific performance is often more critical than aggregate metrics.

Round 2

Reviewer 3 Report

Comments and Suggestions for Authors

I would like to thank the authors for their responses to my comments and suggestions. From my point of view, the paper is well-prepared and suitable for publication